# Epigenetic Regulation in Melanoma: Facts and Hopes

**DOI:** 10.3390/cells10082048

**Published:** 2021-08-11

**Authors:** Emilio Francesco Giunta, Gianluca Arrichiello, Marcello Curvietto, Annalisa Pappalardo, Davide Bosso, Mario Rosanova, Anna Diana, Pasqualina Giordano, Angelica Petrillo, Piera Federico, Teresa Fabozzi, Sara Parola, Vittorio Riccio, Brigitta Mucci, Vito Vanella, Lucia Festino, Bruno Daniele, Paolo Antonio Ascierto, Margaret Ottaviano

**Affiliations:** 1Oncology Unit, Department of Precision Medicine, Università degli Studi della Campania Luigi Vanvitelli, 80131 Naples, Italy; emiliofrancescogiunta@gmail.com (E.F.G.); gluca.arrichiello@gmail.com (G.A.); annalisa.pappalardo88@gmail.com (A.P.); annadiana88@gmail.com (A.D.); angelic.petrillo@gmail.com (A.P.); 2Unit of Melanoma, Cancer Immunotherapy and Development Therapeutics, Istituto Nazionale Tumori IRCCS Fondazione Pascale, 80144 Naples, Italy; curvietto.ma@gmail.com (M.C.); vitovanella1@gmail.com (V.V.); luciafestino1984lf@gmail.com (L.F.); paolo.ascierto@gmail.com (P.A.A.); 3Oncology Unit, Ospedale del Mare, 80147 Naples, Italy; davidebosso84@gmail.com (D.B.); rosanovamario@hotmail.com (M.R.); giopas@email.it (P.G.); pierafederico@yahoo.it (P.F.); fabozzit79@gmail.com (T.F.); b.daniele@libero.it (B.D.); 4Oncology Unit, Department of Clinical Medicine and Surgery, Università degli Studi di Napoli Federico II, 80131 Naples, Italy; saraparola3@gmail.com (S.P.); vittorioriccio1990@gmail.com (V.R.); brigitta.mucci@gmail.com (B.M.); 5CRCTR Coordinating Rare Tumors Reference Center of Campania Region, 80131 Naples, Italy

**Keywords:** melanoma, epigenetics, epigenetic drugs, DNA methylation, chromatin remodeling, non-coding RNA, therapeutic resistance

## Abstract

Cutaneous melanoma is a lethal disease, even when diagnosed in advanced stages. Although recent progress in biology and treatment has dramatically improved survival rates, new therapeutic approaches are still needed. Deregulation of epigenetics, which mainly controls DNA methylation status and chromatin remodeling, is implied not only in cancer initiation and progression, but also in resistance to antitumor drugs. Epigenetics in melanoma has been studied recently in both melanoma preclinical models and patient samples, highlighting its potential role in different phases of melanomagenesis, as well as in resistance to approved drugs such as immune checkpoint inhibitors and MAPK inhibitors. This review summarizes what is currently known about epigenetics in melanoma and dwells on the recognized and potential new targets for testing epigenetic drugs, alone or together with other agents, in advanced melanoma patients.

## 1. Introduction

Cutaneous melanoma is the fifth most common cancer in both sexes and is less frequent, but deadlier, than other skin tumors [1]. In the last decades, advances in understanding the inner mechanisms of melanomagenesis and the consequent discovery of new potential pharmaceutical targets have revolutionized the history of this disease. Firstly, immunotherapy—specifically, the use of immune checkpoint inhibitors (ICIs)—has improved the overall survival of all advanced melanoma patients, independently of genomic mutations [2]. Secondly, the introduction of the association of BRAF and MEK inhibitors—two targeted therapies—in clinical practice has changed the therapeutic scenario for advanced melanoma patients harboring BRAF V600 activating mutations, which confer poorer prognoses compared to those with BRAF V600 wild-type status [3]. Moreover, patients with other molecular subtypes of melanoma have been identified over the years: KIT mutated patients, which could benefit from specific tyrosine kinase inhibitors, ref. [4] and NRAS mutated patients, which, unfortunately, are still lacking specific inhibitors [5].

Despite the undeniable improvement in survival rates that has come about with the use of these drugs, more than half of metastatic patients die from melanoma within 5 years [6]. Resistance to therapy is the main reason for disease progression and, ultimately, death [7]; therefore, delaying or overcoming resistance is an important clinical requirement for advanced melanoma patients.

This review focuses on epigenetics, highlighting old and new evidence regarding the role of DNA methylation and chromatin remodeling in melanoma pathogenesis and the impact of these on new treatment approaches.

## 2. Materials and Methods

An extended review of literature through PubMed was conducted using the keywords “melanoma” and “epigenetic”, also including relevant abstracts from the main societies of oncology around the world and ongoing clinical trials from clinicaltrials.org (accessed on 11 June 2021).

## 3. Epigenetic Regulation and Melanoma

Epigenetics is defined as modifications of DNA molecules or associated factors which, other than the DNA sequence itself, have information content and are preserved during mitosis. There are three major types of epigenetic inheritance that have been identified: DNA cytosine methylation; chromatin remodeling, mostly achieved via histone modification such as acetylation, methylation, and phosphorylation; and non-coding RNA (ncRNA) regulation (Figure 1). In recent years, it has emerged that the role of epigenetic inheritance in many physiological and pathophysiological conditions, including cancer, is involved not only in existing differences between senescent and everlasting cells, but also between somatic and tumor cells [8]. The impact of epigenetics on human tumorigenesis is so decisive that a new classification of cancer genes was recently proposed; the authors identify epigenetic modulators, epigenetic modifiers, and epigenetic mediators, all of which are involved in cancer onset, growth, and progression [9].

In this section we will briefly recapitulate the main mechanisms of epigenetic regulation in human cancer, focusing on their role in the malignant melanoma pathogenesis.

### 3.1. DNA Methylation Status

DNA methylation is a common epigenetic mark resulting from the covalent transfer of a methyl group onto the cytosine of a DNA strand by the activity of DNA methyltransferases (DNMTs) [10]. Interestingly, in somatic cells this physiological event occurs mainly in CpG islands, which are genome regions containing a great number of cytosine-guanine dinucleotide repeats; CpG islands are mainly located on gene promoters, which are the sites of transcription initiation [11].

The first epigenetic deregulation found in tumor cells was DNA methylation loss at CpG dinucleotides [12], with new evidence accumulating in the following years [13,14,15]. Hypomethylation can lead to proto-oncogene activation, chromosomal instability, and drug resistance [16,17,18]. During the initial steps of carcinogenesis, cancer cells frequently display selective hypermethylation of tumor suppressor genes [19], which sometimes defines a CpG island methylator phenotype (CIMP) [20,21]. CIMP was firstly described in colorectal cancer, identifying a tumor subset characterized by epigenetic instability and silencing of several tumor suppressor genes. However, it then became clear that CIMP could also be found in other human cancer types, including melanomas [22].

DNA methylation aberrations are the most frequently described epigenetic alterations in malignant melanoma, particularly when they have been proven to be either pathogenetic or prognostic. Focal DNA hypermethylation of known tumor suppressor genes is a frequent event in melanoma. The phosphatase and TENsin homolog (PTEN) enzyme converts phosphatidylinositol 3,4,5-triphosphate (PIP3) into phosphatidylinositol 4,5-biphosphate (PIP2), antagonizing the phosphoinositide 3-kinase (PI3K) function and thus suppressing the activation of the PI3K/AKT pathway; functional inactivation of PTEN by deletion or mutation occurs in 30–60% of sporadic melanomas [23]. PTEN promoter methylation was reported in cell-free DNA from 62% of the melanoma serum samples examined by pyrosequencing, indicating a good correlation with the same epigenetic alteration found in paired melanoma tissues, and investigated through reverse transcriptase polymerase chain reaction (PCR) [24]. Similar results were obtained using methylation-specific PCR in a different series, in which, moreover, PTEN promoter methylation was identified as an independent predictor of poor outcome [25]. The p16INK4a protein, encoded by the CDKN2A gene (locus 9p21), plays an important role in regulating cell cycle phase transitions; several studies performing analyses on melanoma samples reported a methylation frequency in the CDKN2A promoter ranging from 5% to 27% and a significant overrepresentation in NRAS-mutated samples [26,27,28,29]. On the same locus, the ARF gene encodes for a different tumor suppressor protein, p14ARF, which was silenced—through promoter hypermethylation—in up to 57% of melanoma samples examined in the study by Freedberg and colleagues, independently of CDKN2A promoter methylation status [30]. Finally, the tumor suppressor gene RAS association domain family protein 1 (RASSF1A), encoding for a microtubule-associated protein involved in cell cycle regulation and apoptosis, was reported as being hypermethylated in 55% of melanoma tumors [31] (Figure 1A). Apart from known tumor suppressors, over one hundred other genes belonging to cancer cell survival and growth pathways have been found to be differentially methylated in melanoma as a result of methylome-wide analysis in both cell lines and clinical samples [32,33,34,35,36]. These features were also able to discriminate between benign melanocytic nevi and malignant melanoma lesions.

Moreover, genome-wide methylation studies recently provided insights into resistance to immunotherapeutic strategies. Indeed, hypermethylation may limit the efficacy of immune checkpoint blockade therapy by inhibiting cancer cell recognition through the suppression of endogenous interferon responses. It was recently observed that hypermethylation of cGAS and stimulator of interferon genes (STING), both involved in innate immunity, contributes to the disruption of the STING signaling function and reduces tumor antigenicity, fueling melanoma resistance to T-cell-based anticancer therapies [37]. On the other hand, global hypomethylation could increase expression of programmed death ligand 1 (PD-L1) and inhibitory cytokines, which would definitely contribute to immunosuppression. Furthermore, it has been suggested that DNA methylation plays a key role in cytotoxic T-cell ‘exhaustion’, a phenomenon associated with tumor progression [38]. Finally, PD-L1, programmed death ligand 2 (PD-L2), and cytotoxic T-lymphocyte antigen 4 (CTLA-4) gene methylation status—and, consequently, their expression levels—have displayed prognostic significance and predictive value in patients treated with ICIs [39,40,41].

Apart from the deregulated promoter methylation status, melanoma also exhibits global hypomethylation within its genome; loss of immunohistochemistry staining for 5-hydroxymethylcytosine (5-hmC), which is one of the most abundant metabolites resulting from active DNA demethylation, can help in differentiating malignant melanomas from benign melanocytic proliferations. The progressive loss of 5-hmC staining has been shown to correlate with some parameters which predict shorter recurrence-free survival and overall survival [42]. It has been suggested that this feature could be attributed to isocitrate dehydrogenase 2 (IDH-2) downregulation or mutation, disrupting ten eleven translocase (TET) ability to maintain DNA methylation fidelity [43].

The source of aberrant DNA methylation in melanoma remains elusive, but mounting evidence suggests that it might depend on deregulated proliferative signaling pathways, specifically PI3K/AKT and MAPK [44] (Table 1).

### 3.2. Chromatin Remodeling

Chromosomal DNA is packed into nucleosomes with DNA enveloped around histone protein complexes, which consist of subunits named H2A, H2B, H3, and H4, whose modification can either activate or silence gene transcription. A recent paper divided such modifications into “writers”, “readers”, and “erasers”, according to their role in epigenetic regulation of gene expression [63]. Among the most studied alterations, histone subunit methylation and acetylation can regulate genetic expression by controlling DNA accessibility to the transcriptional machinery and also through the participation of additional protein complexes [64]. For example, histone H3 lysine 4 di/tri-methylation (H3K4me2/3) and histone H3 acetylation (H3ac) are generally associated with gene hyperexpression, whereas H3 lysine 27 methylation (H3K27me) is associated with gene inactivation; histone modifications in cancer usually determine the silencing of tumor suppressor genes and are generally associated with a bad prognosis [65,66].

Although histone alterations are the least documented epigenetic mechanisms involved in the melanomagenesis, it has been suggested that both aberrant deacetylation and methylation play an active role in this process. Histone hypoacetylation is associated with downregulation of proapoptotic proteins, including antiapoptotic members of the Bcl-2 family and PI3K/AKT signaling pathway regulators [45,67]. Hypermethylation mediated by two histone-lysine methyltransferases, enhancer of zeste homolog 2 (EZH2)—member of the polycomb repressor complex 2 (PRC2)—and SET domain bifurcate 1 (SETDB1), can cause deregulation of genes associated with cellular development and cell cycle progression [46,47] (Figure 1B). Interestingly, EZH2 seems to play a role in melanomagenesis, given that its expression increases from benign melanocytic proliferations to malignant melanoma [68]. Similarly, the euchromatic histone-lysine methyltransferase 2 (EHMT2) can drive melanoma growth and promote an immunosuppressive microenvironment by activating the WNT signaling pathway [48]. A different methyltransferase, known as histone-lysine N-methyltransferase 2D (KMT2D), might play a role in deregulating genes which are critical for cell migration and actively participate in melanomagenesis, as shown in a study using patient derived tumor samples [49].

Members of the bromodomain and extra-terminal motif (BET) protein family, such as bromodomain-containing proteins 2 (BRD2) and 4 (BRD4), which are able to recognize specific chromatin modifications and subsequently initiate downstream regulatory processes, are key controllers of cell cycle and survival gene expression and have been found to be overexpressed in melanoma cells. Intriguingly, in vitro inhibition of such “chromatin readers” is associated with cell cycle arrest in preclinical melanoma models, regardless of BRAF or NRAS mutational status [51,52,53].

Histone demethylase deregulation resulting in abnormal histone methylation patterns has also been related to melanomagenesis; more specifically, expression of histone demethylase JARID1B has been implied in the self-renewal capability of a slow-cycling melanoma subpopulation responsible for tumor growth and, possibly, for intrinsic drug resistance [50].

Recently, a new model of tumorigenesis has emerged in which several stress factors, such as pH, radiation, and hypoxia, allow malignant cells to revert to a stem cell-like phenotype through microenvironment-mediated epigenetic regulation of gene expression [69,70,71].

Further evidence was recently published on this matter using melanoma models. Through an in vitro tumor, microengineering approach, authors demonstrated that stress exerted on melanoma aggregate perimeters could determine the cellular phenotypic transition into a stem cell-like melanoma-initiating cell (MIC) state. In this work, H3K9ac and H3K4me2 histone modifications and the enrichment of epigenetic modifier PR domain zinc finger 14 (PRDM14) were correlated to stemness, suggesting their role in cellular plasticity and tumorigenicity [72].

Chromatin structure can also be changed through the ATP-dependent remodeling activity of the SWItch/sucrose non-fermentable (SWI/SNF) complexes: there are three known complexes—named canonical BAF (cBAF), polybromo-associated BAF (PBAF), and noncanonical BAF (ncBAF)—made of subunits which are encoded by 29 genes [73]. These complexes facilitate the repair of DNA double strand breaks and UV-induced pyrimidine dimers in order to maintain genomic stability and avoid mutations and structural aberrations in the chromosomes [74,75,76]. Loss-of-function mutations in the SWI/SNF complex components such as AT-rich interactive domain-containing protein 1A (ARID1A), ARID1B, ARID2, or SMARCA4 are frequent in melanoma, suggesting that altered chromatin remodeling plays a role in the pathogenesis of this disease [54]. Loss of ATRX, a member of the same family, has also been linked to melanoma progression [55]. Recently, a different chromatin remodeling complex has been implicated in the pathogenesis of melanoma: the nucleosome remodeling factor (NURF) complex, specifically its bromodomain PHD finger transcription factor (BPTF) subunit, participates in the proper functioning of microphthalmia-associated transcription factor (MITF). In detail, MITF regulates melanocyte physiology and subsequently dictates fundamental gene expression programs in melanoma [56].

Histone variants, which are another mechanism of chromatin alteration, replace canonical histones in defined regions of the genome and could, therefore, modify chromatin structure and gene expression. Loss of histone variant macroH2A, which is generally considered to be transcriptionally repressive, promotes melanoma progression [77]; on the other hand, overexpression of histone variant H2A.Z.2 promotes cell cycle progression and is associated with poor prognoses in melanoma [78] (Table 1).

### 3.3. Non-Coding RNA Regulation

NcRNAs are a heterogeneous group of RNAs that are generally divided into two bands, small or long, based on their length [79]. MicroRNAs (miRNAs) with a length of 20–25 nucleotides are the best studied small ncRNAs; they regulate translation through the binding of specific response elements included in their target mRNA transcripts and, subsequently, recruit RNA-induced silencing complexes which antagonize target mRNA stability and/or translation [80]. Long non-coding RNAs (lncRNAs), whose length ranges from 200 nucleotides to 100 kbs, could perform several gene regulatory roles, e.g., chromosome dosage compensation, genomic imprinting, and epigenetic regulation, among others [81] (Figure 1C).

The prognostic and pathobiological importance of ncRNAs in melanoma has been well established. MiR-200c is significantly more downregulated in both primary and metastatic melanoma samples compared with benign melanocytic nevi, causing reduced expression of adhesion molecules, such as E-Cadherin, and transcriptional repression of the CDKN2A locus [57]. On the other hand, several miRNAs have been found to exhibit oncogenic or pro-metastatic capabilities in melanoma. Mi-R-149, miRNA 21, and several other miRNAs clustered on the X chromosome are associated with apoptosis inhibition and are overexpressed in melanoma samples [58,59,82]. In addition, higher expression of a cooperative network of miRNAs targeting apolipoprotein E (miRNA-1908, miR-199a-5p, and miR-199a-3p) appears to promote invasion and metastasis and is associated with shorter metastasis-free survival [60]. MiRNAs are also implicated in primary and metastatic melanoma niche modulation, regulating tumor-specific immune responses and mediating treatment resistance [83].

HOTAIR, a lncRNA, was found to be overexpressed in lymph nodal melanoma metastasis and was suggested as interacting directly with histone-modifying enzymes, thus altering chromatin structure [61]. Additional putative oncogenic lncRNAs have been reported in melanoma, such as MALAT1, BANCR, ANRIL, SPRY-IT1, and SAMMSON, which seem to participate in melanomagenesis through several mechanisms such as apoptosis inhibition, invasion, and metastasis formation [62] (Table 1).

## 4. Targeting Epigenetic Machinery in Melanoma

After their discovery, epigenetic alterations rapidly became new potential therapeutic targets in human diseases and particularly in human cancers [84]. Indeed, unlike genomic mutations, epigenetic alterations in cancer are, at least in principle, therapeutically reversible, thus attracting increasing attention for drug development in recent decades [85].

Several epigenetic therapies based on the inhibition of histone deacetylases (HDAC-i) and DNA methyltransferases (DNMT-i) have already received Food and Drug Administration (FDA) approval, specifically for the treatment of myelodysplastic syndromes and cutaneous T-cell lymphomas.

Concerning melanoma, preclinical studies testing epigenetic drugs, such as the hypomethylating agent 5-azacytidine and the pan-HDAC-i panobinostat, showed interesting antitumor effects [86,87]. Unfortunately, the use of single-agent, first-generation epigenetic drugs in melanoma patients did not translate into meaningful clinical activity, probably due to a lack of selectivity [88]. Therefore, research is currently focused on combining epigenetic drugs with existing immunotherapeutic, chemotherapeutic, and radiotherapeutic approaches to enhance their efficacy and tackle potential resistance to treatments [89]. The following section will focus on known epigenetic mechanisms of resistance to currently available treatments and highlight ongoing efforts to overcome this phenomenon.

### 4.1. Immunotherapy

ICIs have revolutionized the therapeutic scenario of melanoma patients, the current standard treatment in advanced stages being antibodies against programmed death-1 (PD-1) (i.e., pembrolizumab, nivolumab) and CTLA-4 (i.e., ipilimumab) [90]. However, the onset of both primary and acquired resistance to ICIs affects more than a half of the patients receiving these agents in the first-line setting [91].

To date, only a few studies have described epigenetic mechanisms of resistance to immunotherapeutic agents in melanoma.

The histone methyltransferase EZH2 plays a fundamental role in modulating T-cell responses, promoting survival, and the function of both CD4+ helper and CD8+ cytotoxic T-cells [92]. Elevated expression of EZH2 was already shown to be associated with poor prognoses in melanoma patients [93]. Moreover, a study highlighted that EZH2 is upregulated in melanoma murine models upon treatment with anti-CTLA-4 and IL-2 immunotherapies, with subsequent repression of critical immune-related genes in tumor infiltrating lymphocytes such as PD-L1, T-cell immunoglobulin and mucin domain-3 (TIM-3), and lymphocyte activation gene-3 (*LAG-3*) [94]. Furthermore, the authors of this study demonstrated that EZH2 inhibition could restore tumor immunogenicity and T-cell infiltration and suppress melanoma growth upon re-challenging with immunotherapy.

Another potential mechanism of resistance to anti-PD-1 therapy could be the histone H3K4 demethylase LSD1, whose depletion renders refractory murine models of tumors responsive to ICIs through activation of type 1 interferon, as a result of induced double strand RNA stress [95].

A genome-scale CRISPR-Cas9 screen has identified the inactivation of ARID2, BRG1, and BRD7—all of which are members of the PBAF form of the SWI/SNF complex—to be responsible for the sensitization of melanoma murine models to cytotoxic T-cells via an enhanced response to IFN-gamma [96].

All of the above-mentioned evidence provided the rationale for combining epigenetic drugs and ICIs in melanoma patients; moreover, in the preclinical setting, the combination of HDAC-i and immunotherapeutic agents—or adoptive T-cell transfer—has yielded promising results in melanoma models [97,98,99]. Based on these early results, a few clinical trials were launched which investigated the combination of HDAC-i and ICIs; available results have been published in abstract form only. A phase I trial investing a combination of panobinostat—a pan inhibitor of class I, II, and IV histone deacetylases—and ipilimumab in advanced pre-treated melanoma did not show an increased response compared to standard ipilimumab alone, with several dose-limiting toxicities related to panobinostat when administered at higher doses [100]. The authors suggested studying more selective HDAC-I for the next clinical trials. In the phase Ib/II trial, SENSITIZE, advanced stage melanoma patients who progressed to prior checkpoint inhibitor therapy were treated with the selective class I HDAC-i domatinostat together with pembrolizumab. The combination was found to be safe and tolerable with potential antitumor efficacy, due to an increase in CD8+ T-cells infiltration and changes in the immune tumor microenvironment [101]. Finally, the ENCORE-601 trial evaluated a combination of pembrolizumab and entinostat, a class I histone deacetylase inhibitor, in patients with melanoma previously treated with and anti-PD1 therapy [102]. Following the combinatorial treatment, of the fifty-three evaluable patients, a partial response and a complete response was reached in nine patients and one patient, respectively, giving an objective response rate of 19%. At the time of data cut-off, the median duration of response was 12.5 months with five patients experiencing ongoing responses. Grade 3/4 related adverse events (AEs) occurring in >5% of patients included neutropenia, fatigue, and hyponatremia. Five patients (9%) experienced a Grade 3/4 immune-related AE (two events of rash, one each of colitis, pneumonitis, and autoimmune hepatitis).

Ongoing trials are currently exploring different combinations, including pembrolizumab plus entinostat (NCT03765229); pembrolizumab plus oral azacytidine (NCT02816021); nivolumab plus tinostamustine, a fusion molecule composed of the alkylating agent bendamustine fused to the pan-HDACi vorinostat (NCT03903458); ipilimumab plus SGI-110, a precursor of the DNMT-i decitabine (NCT02608437); ipilimumab and nivolumab plus ACY-241, a selective HDAC6 inhibitor (NCT02935790). Unfortunately, no updated results from these trials have been posted yet (Table 2).

### 4.2. Targeted Therapy

BRAF codon V600 activating mutations, which affect almost a half of melanoma patients, became a therapeutic target with the development of specific inhibitors—namely BRAF and MEK inhibitors—which are currently used in both adjuvant and metastatic settings [108]. Despite the substantial advances in the treatment of BRAF-mutated melanoma, emergence of resistance remains a major challenge to the lasting success of targeted therapies. It is interesting to note that in about 40% of the cases resistance to BRAF and MEK inhibitors is not associated with mutational events but rather with epigenetic events, such as differential methylation of CpG sites [109]. It is likely that adaptation processes involve transient alterations in the epigenome, which, therefore, become stable upon continuous drug exposure. Starting from the assumption that epigenetic changes are reversible, this phenomenon is quite relevant for drug resistance as it might explain the significant responses observed in patients that are re-treated with the same drugs after a treatment break [110]. In support of this hypothesis, authors of a recent study have tracked thousands of single melanoma cells over the first 4 days of treatment with the BRAF inhibitor dabrafenib: most cells responded to dabrafenib and became quiescent, whilst a subpopulation of cells was seen escaping drug action within 3 days and re-entering the cell cycle through the ATF4-induced stress response. However, cells that have escaped drug treatment could rapidly revert to the parental drug-sensitive state when the drug is withdrawn, ultimately implicating a non-genetic mechanism of acquired resistance [111]. In BRAF mutant melanoma, resistance to BRAF and MEK kinase inhibitors has also been associated with phenotype plasticity and gene expression program alterations, which identify three different cellular states: a pigmented melanocytic state associated with transcription factor MITF expression; a neural crest-like state characterized by nerve growth factor receptor (NGFR) activation; and an undifferentiated state characterized by low levels of the transcription factor SOX10 and high levels of receptor tyrosine kinases, such as AXL and the epidermal growth factor receptor EGFR [112]. Recently, a library of small molecules acting as epigenetic modulators was used to identify the regulators of the abovementioned plasticity in BRAF-mutant melanoma cell lines in order to explain cell-to-cell variability, despite MAPK dependency. Results from this work have allowed differentiation among three further states: a lysine demethylase 1A (KDM1A)-dependent state, that could be efficiently inhibited by SP2509, a reversible KDM1A inhibitor mainly found in undifferentiated cells; a lysine demethylase 4B (KDM4B)-dependent state that is sensitive to JIB-04, a pan-inhibitor of Jumonji histone demethylases observed in neural crest-like cells; and a state induced by birabresib, which is a BET bromodomain inhibitor [113].

Different epigenetic mechanisms of acquired resistance to MAPK inhibition include global DNA hypomethylation due to differential expression of DNA methyltransferases, elevated expression of histone demethylases, such as JARID1A and JARID1B, and methyltransferases, including SETDB1 and SETDB2 [114,115]. It is worth it noting that some epigenetic alterations described in this work, such as loss of H3K4me3 and H3K27me3 or gain of H3K9me3, are associated with multidrug resistance (so called induced drug-tolerant cells, IDTCs), a phenomenon which seems to be reversible after drug holidays [116]. One of the most recently uncovered resistance mechanisms involves haploinsufficiency, but not complete loss, of the histone deacetylase Sirtuin6 (SIRT6). The haploinsufficiency of this enzyme determines an increase in insulin-like growth factor binding protein 2 (IGFBP2) expression and the subsequent activation of the insulin-like growth factor receptor (IGF-R) signaling, which cause resistance to BRAF and MEK inhibition in BRAF-mutant melanoma models [117].

Several strategies which adopt epigenetic drugs are currently being explored to overcome targeted therapy resistance in melanoma. A mechanism of acquired resistance in BRAF-mutant melanoma involves overexpression of YAP, a component of the Hippo pathway, whose upregulation is mediated by BET bromodomain proteins. In BRAF V600E melanomas, combination treatment inhibiting BET bromodomains and either BRAF V600E or MEK kinases synergistically block cell proliferation [103,118]. Acquired resistance to BRAF/MEK inhibitors seems to cause increased production of reactive oxygen species (ROS) in BRAF V600E mutant melanoma cells; this production could be exacerbated and exploited by therapeutic inhibition of HDAC and subsequent downregulation of the SLC7A11 transporter [119]. In preclinical models of BRAF-mutant melanoma, the aforementioned selective class I HDAC-i entinostat was not able to suppress tumor growth; however, when simultaneously combined with MAPK inhibitors, a regression of 70% was observed [104]. Finally, it has been suggested that pharmacological inhibition of histone H3K9 demethylase, either alone or in combination with MAPKi, and co-targeting of EZH2 together with one of its regulators, NFATc2, may benefit the treatment of resistant BRAF V600E melanomas [120,121].

In a clinical setting, a combination of the hypomethylating agent decitabine with the BRAF inhibitor vemurafenib was found to be safe and effective in both treatment-naïve and pre-treated BRAF-mutant melanoma patients [105]; a follow-up study evaluating the addition of cobimetinib to this combination was unfortunately terminated due to loss of funding (NCT01876641) (Table 2).

### 4.3. Chemotherapy/Radiotherapy

Aside from possible combinations with either immunotherapy or targeted agents, epigenetic modulators may also enhance the effectiveness of standard chemotherapeutic or radiotherapeutic regimens [122]. Combinations of decitabine plus temozolomide and decitabine/panobinostat plus temozolomide have been found to be safe and effective in treating metastatic melanoma; the synergistic effect might be explained by the restoration of aberrant methylation patterns linked with O-6-methylguanine-DNA methyltransferase (MGMT) hyperactivity [106,123]. Furthermore, DNMT and HDAC inhibitors can restore apoptotic capacity by upregulating epigenetically silenced effectors such as Apaf-1, caspase-8, and p16, therefore enhancing the chemosensitivity of melanoma cells to doxorubicin, cisplatin, and etoposide [124,125,126]. In particular, the anticonvulsant valproic acid, which displays histone deacetylase inhibiting activity, was found to be effective when combined with the topoisomerase inhibitor karenitecin in a phase I/II trial [107] (Table 2).

The intriguing possibility of exerting epigenetic changes—especially DNA methylation and histone modifications—through exposure to ionizing radiation (IR), as shown in several studies, has become a field of interest in cancer research [127]. It has also been suggested that epigenetic remodeling may tune the radiosensitivity of cancer cells; in fact, HDAC inhibitors, given their demonstrated ability to restore the apoptosome in melanoma, may radiosensitize human melanoma cells, as shown in pre-clinical models [128,129].

## 5. Future Directions and Conclusions

Advanced melanoma is, in most cases, a lethal disease for which new therapeutic strategies are needed. Epigenetics has emerged as an intriguing field of research in human cancer, but more efforts should be made to understand how to take advantage of it.

Many of the epigenetic drugs mentioned in this review have demonstrated safe initial signs of efficacy in melanoma patients, but several questions need to be answered: Is it possible to predict a response from epigenetic drugs by using biomarkers, as currently happens for targeted therapies? Is the timing of epigenetic drug use crucial for their efficacy? Should we think about alternative schedules of administration—i.e., on/off treatment periods, pre-planned dosage variations—to achieve the best possible result?

Another important chapter in melanoma research is the treatment of brain metastases, which occur in up to 75% of metastatic melanoma patients during the course of the disease and confer poor prognoses, despite targeted therapy and immunotherapy [130]. Epigenetic regulation could, therefore, be an intriguing therapeutic target for patients with synchronous or metachronous brain metastases from melanoma, given the high percentage of epigenetic alterations found in brain specimens [131]. Moreover, a recent work has identified a specific epigenetic signature for melanoma brain metastases through DNA methylation analysis [132]. Despite this, clinical trials investigating the role of epigenetic drugs in this specific subset of patients are lacking.

A new chapter in epigenetic research is currently focusing on tumor microenvironments, given their close association with tumor cells and their influence on tumor growth and immune-regulation [133]. Concerning melanoma, a recent work using CpG methylation analysis has evaluated three immune methylation clusters in tumor-infiltrating immune cells, which are associated with the survival of melanoma patients, suggesting a correlation between epigenetic immune regulation and prognosis [134]. Epigenetic modulation of tumor microenvironments could, therefore, be an optimal therapeutic target for tumors which typically exhibit a considerable immune infiltrate, as is the case with melanoma.

In conclusion, ongoing clinical trials will certainly add more information and optimize epigenetic drug use in melanoma patients, even though the unique characteristics of epigenetic regulation in somatic and cancer cells and epigenomic plasticity should always be considered when ideating preclinical and clinical projects.

## Figures and Tables

**Figure 1 cells-10-02048-f001:**
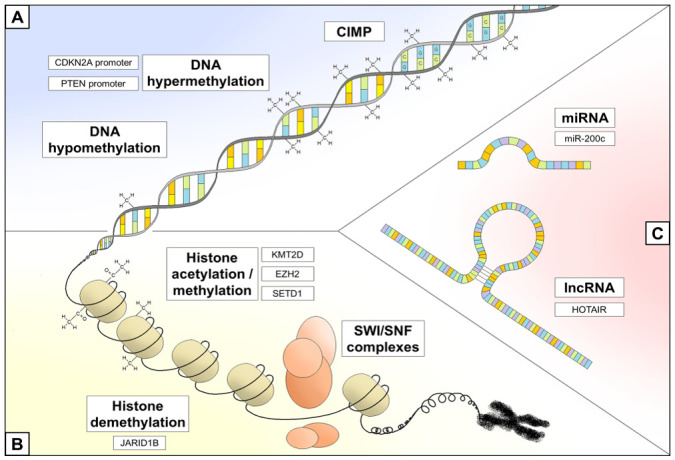
Main epigenetic mechanisms involved in melanoma onset and progression (bold text), with some examples of alterations for each mechanism. (**A**) DNA methylation status; (**B**) chromatin remodeling; (**C**) non-coding RNA. Abbreviations: CIMP, CpG island methylator phenotype; miRNA, microRNA; lncRNA, long non-coding RNA; SWI/SNF, SWItch/sucrose non-fermentable.

**Table 1 cells-10-02048-t001:** Main epigenetic mechanisms altered in melanoma. Abbreviations: miRNA, micro-RNA; lncRNA, long non-coding RNA.

Class of Epigenetic Alteration	Type of Epigenetic Alteration	Involved Gene(s)	Evidence in Melanoma	Ref
DNA Methylation status modification	Promoter methylation	PTEN	Up to 60% of melanoma serum samples Independent poor prognostic factor	[24,25]
		CDKN2A	Up to 25% of melanoma tissue samples Cell cycle deregulation	[26,27,28]
		ARF	Up to 60% of melanoma tissue samples Cell cycle deregulation	[30]
	Gene methylation	RASSF1A	Up to 55% of melanoma tissue samples	[31]
		cGAS and STING	Resistance to T-cell-based anticancer therapies	[37]
	Gene hypomethylation	IDH-2 (?)	Disruption of TET ability to maintain DNA methylation fidelity	[43]
Chromatin remodeling perturbation	Histone hypoacetylation	Bcl2	Downregulation of antiapoptotic members	[45]
	Histone hypermethylation	EZH2	High proliferation rate and aggressive tumor subgroups	[46]
		SETDB1	Acceleration of melanoma onset	[47]
		EHMT2	Up to 25% of melanoma tissue samples	[48]
		KMT2D	Participation to melanomagenesis	[49]
		JARID1B	Tumor growth and intrinsic drug resistance	[50]
	Chromatin modification recognition	BRD2 and BRD4	Apoptosis inhibition and cell cycle deregulation	[51,52,53]
	SWI/SNF complex regulation	ARID1A, ARID1B, ARID2, and SMARCA4	Loss of ability to repair DNA double strand breaks and UV-induced pyrimidine dimers	[54]
		ATRX	Melanoma progression	[55]
	NURF complex regulation	BPTF	Disruption of gene expression programs	[56]
Non-coding RNA regulation	miRNA	MiR-200c	Reduced expression of adhesion molecules	[57]
		MiR-149 and MiR-21	Apoptosis inhibition	[58,59]
		MiR-1908, miR-199a-5p, and miR-199a-3p	Promotion of invasion and metastasis formation Shorter overall survival	[60]
	lncRNA	HOTAIR	Alteration of chromatin structure	[61]
		MALAT1	Apoptosis inhibition, promotion of invasion and metastasis formation	[62]

**Table 2 cells-10-02048-t002:** Association of epigenetic drug with other therapeutic classes in preclinical melanoma models and melanoma patients. Abbreviations: HDACi, histone deacetylase inhibitor; DNMTi, DNA methyltransferase inhibitor.

Class of Therapeutic Partner	Partner Drug	Epigenetic Drug	Evidence in Melanoma	Ref
Immunotherapy	9H10 (anti-CTLA-4 antibody)	5-aza-2′-deoxycytidine (DNA hypomethylating agent)	Significant immune-related antitumor activity in syngeneic transplantable murine models	[97]
	Gp100 melanoma antigen-specific pmel-1 T-cells (adoptive transfer)	LAQ824 (HDACi)	Improvement of antitumor activity in murine models	[98]
	Ipilimumab (anti-CTLA-4 antibody)	Panobinostat (pan-HDACi)	Phase I clinical trial: not increased responses in advanced melanoma patients respect to single-agent ipilimumab	[100]
		SGI-110: precursor of decitabine (DNMTi)	Phase 1 clinical trial (ongoing)	NCT02608437
	Pembrolizumab (anti-PD-1 antibody)	Domatinostat (class I HDACi)	Phase Ib/II clinical trial (SENSITIZE): safety and tolerability of combination and potential increase in antitumor activity in pretreated melanoma patients	[101]
		Entinostat (class I HDACi)	Ph1b/2 Dose-Escalation Study (ENCORE 601) of Entinostat with Pembrolizumab in NSCLC with Expansion Cohorts in NSCLC, Melanoma, and Colorectal Cancer	[102]
		Entinostat (class I HDACi)	Phase II clinical trial (ongoing)	NCT03765229
		Azacytidine	Phase II clinical trial (ongoing)	NCT02816021
	Nivolumab (anti-PD-1 antibody)	Tinostamustine: fusion molecule of bendamustine (alkylating agent) + vorinostat (pan-HDACi)	Phase I clinical trial (ongoing)	NCT03903458
	Ipilimumab (anti-CTLA-4 antibody) + nivolumab (anti-PD-1 antibody)	ACY-241 (HDAC6 inhibitor)	Phase I clinical trial (ongoing)	NCT02935790
Targeted therapy	PD901 (MEK inhibitor)	JQ-1 (BET inhibitor)	Reversion of therapeutic resistance in preclinical models	[103]
	Dabrafenib (BRAF inhibitor) + trametinib (MEK inhibitor)	Vorinostat (pan-HDACi)	Enhancement of tumor regression	[104]
	Vemurafenib (BRAF inhibitor)	Decitabine (DNMTi)	Phase I clinical trial (interrupted)	[105]
Chemotherapy	Temozolomide (alkylating agent)	Decitabine (DNMTi)	Phase I/II clinical trial	[106]
	Karenitecin (topoisomerase 1 inhibitor)	Valproic acid (HDACi)	Translational study Phase I/II clinical trial	[107]

## Data Availability

The corresponding author will provide the data, or will cooperate fully in obtaining and providing the data, on which the manuscript is based for examination by the editors or their assignees.

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
