# Peer review of "Epigenetic Regulation in Melanoma: Facts and Hopes"

_cells, 2021, doi:10.3390/cells10082048_

Round 1

Reviewer 1 Report

A very well-performed review about the current understanding of the epigenetic role in cutaneous melanoma and its possible uses in the therapy of this type of cancer.

The review is very clear and coincise.

I would probably just add a materials and methods section indicating what keywords you used and what databases you searched (Pubmed, Google Scholar, Scopus...etc...) in order to select the studies. Also, there are too many authors for such a small work (20 authors are too many). 

I would probably also change the title from "Epigenetic Regulation in Cutaneous Melanoma: Facts and  Hopes" to "Epigenetic Regulation in Advanced Melanoma: state of the art", as melanomas are not limited to the skin and all that is reported in the paper is basically applicable to advanced melanomas ( in situ melanoma are usually only managed with surgery and no other treatments are needed).

Thank You

Author Response

A very well-performed review about the current understanding of the epigenetic role in cutaneous melanoma and its possible uses in the therapy of this type of cancer. The review is very clear and concise.

  • I would probably just add a materials and methods section indicating what keywords you used and what databases you searched (PubMed, Google Scholar, Scopus...etc...) in order to select the studies.

Authors: We greatly appreciate the First Reviewer's valuable comments. According to her/his suggestions the paragraph number 2 has been added to describe the methods we used.

  • Also, there are too many authors for such a small work (20 authors are too many).

Authors: We thank the reviewer for the careful observation, however this manuscript is on behalf of the SCITO YOUTH (Società Campana di Immunoterapia Oncologica), which is a collaborative group of young Italian researchers who collaborate all together on writing projects. For this reason, we would prefer not change the authors ‘list.

  • I would probably also change the title from "Epigenetic Regulation in Cutaneous Melanoma: Facts and Hopes" to "Epigenetic Regulation in Advanced Melanoma: state of the art", as melanomas are not limited to the skin and all that is reported in the paper is basically applicable to advanced melanomas (in situ melanoma are usually only managed with surgery and no other treatments are needed).

Authors: We appreciate the reviewer's suggestion and we eliminated the term “cutaneous” from the title, but we would like to maintain “facts and hopes” since many things need to be elucidated in the future and the hope is that coming discoveries will improve the treatment of these patients.

Reviewer 2 Report

Giunta et al. present here a very extensive and comprehensive literature review of the epigenetic regulation in cutaneous melanoma. The last chapter focuses on the clinical applications of targeting specific epigenetic regulators combining targeted inhibitors with current chemo/immunotherapies.

Very mild comments:

  • Some references are old and would benefit the addition of newer studies.
  • 271-273: Therefore, research is currently focused on combining epigenetic drugs with existing immunotherapeutic, chemotherapeutic and radiotherapeutic approaches, to enhance their efficacy and tackle potential resistance to treatment.

Some references are necessary here.

  • There are no mention of Melanoma brain metastasis which occur in 50% of melanoma patients… any correlation with the epigenetic regulation could be described?

Author Response

Giunta et al. present here a very extensive and comprehensive literature review of the epigenetic regulation in cutaneous melanoma. The last chapter focuses on the clinical applications of targeting specific epigenetic regulators combining targeted inhibitors with current chemo/immunotherapies.

Very mild comments:

  • Some references are old and would benefit the addition of newer studies.

Authors: We greatly appreciate the Second Reviewer's valuable comments, however since the manuscript is focused on an evolving topic, we thought to highlight in the text the most relevant and pioneering studies, leaving the most recent ones with the NCT reference.

  • 271-273: Therefore, research is currently focused on combining epigenetic drugs with existing immunotherapeutic, chemotherapeutic and radiotherapeutic approaches, to enhance their efficacy and tackle potential resistance to treatment. Some references are necessary here.

Authors: We thank the reviewer for this careful observation and accordingly we added a reference (Cossio et al 2020, number 89) which summarize this topic; however, this period is only a brief summary of what is explained after in the same paragraph.

  • There is no mention of Melanoma brain metastasis which occur in 50% of melanoma patients… any correlation with the epigenetic regulation could be described?

Authors: We apologize for the lack of mention of melanoma brain metastasis. We completely share the reviewer's comment and we have added a small paragraph in the last chapter about this topic. Unfortunately, to date, there is small data about epigenetic in this subset of melanoma patients, so it remains an open issue.